# Prevalent and Drug-Resistant Phenotypes and Genotypes of *Escherichia coli* Isolated from Healthy Cow’s Milk of Large-Scale Dairy Farms in China

**DOI:** 10.3390/ijms26020454

**Published:** 2025-01-08

**Authors:** Jiaojiao Gao, Yating Wu, Xianlan Ma, Xiaowei Xu, Aliya Tuerdi, Wei Shao, Nan Zheng, Yankun Zhao

**Affiliations:** 1Ministry of Agriculture and Rural Affairs-Laboratory of Quality and Safety Risk Assessment for Agro-Products, Key Laboratory of Agro-Products Quality and Safety of Xinjiang, Institute of Quality Standards & Testing Technology for Agro-Products, Xinjiang Academy of Agricultural Sciences, Urumqi 830091, China; 320222682@stu.xjau.edu.cn (J.G.); yatingwu@xaas.ac.cn (Y.W.); 320242735@stu.xjau.edu.cn (X.M.); 320232680@stu.xjau.edu.cn (X.X.); 320222697@stu.xjau.edu.cn (A.T.); 2Xinjiang Meat and Milk Herbivore Nutrition Laboratory, College of Animal Science Xinjiang Agriculture University, Urumqi 830052, China; dksw@xjau.edu.cn; 3Key Laboratory for Quality and Safety Control for Milk and Dairy Products of Ministry of Agriculture and Rural Affairs, Institute of Animal Science, Chinese Academy of Agricultural Sciences, Beijing 100193, China; zhengnan_1980@126.com

**Keywords:** *Escherichia coli*, antimicrobial resistance, antimicrobial resistance, drug resistance gene, raw milk

## Abstract

*Escherichia coli* is a common cause of mastitis in dairy cows, which results in large economic losses to the livestock industry. The aim of this study was to investigate the prevalence of *E. coli* in raw milk in China, assess antimicrobial drug susceptibility, and identify key antibiotic resistance genes carried by the isolates. In total, 350 raw milk samples were collected from large-scale farms in 16 provinces and cities in six regions of China to assess the resistance of *E*. *coli* isolates to 14 antimicrobial drugs. Among the isolates, nine resistance genes were detected. Of 81 *E. coli* isolates (23.1%) from 350 raw milk samples, 27 (33.3%) were multidrug resistant. Antimicrobial susceptibility testing showed that the 81 *E. coli* isolates were resistant to 13 (92.9%) of the 14 antibiotics, but not meropenem. The resistance gene *blaTEM* was highly distributed among the 27 multidrug-resistant isolates with a detection rate of 92.6%. All isolates carried at least one resistance gene, and 19 patterns of resistance gene combinations with different numbers of genes were identified. The most common gene combinations were the one-gene pattern *blaTEM* and the three-gene pattern *blaTEM-blaPSE-blaOXA*. The isolation rate of *E. coli* in raw milk and the identified resistance genes provide a theoretical basis for the rational use of antibiotics by clinical veterinarians.

## 1. Introduction

*Escherichia coli* is a Gram-negative, opportunistic pathogen commonly found in natural environments and a major cause of persistent and recurrent mastitis in dairy cows [1]. *E. coli* present in raw milk and dairy products can lead to serious foodborne illnesses in humans, including hemolytic uremic syndrome, thrombotic thrombocytopenic purpura, hemorrhagic colitis, and bloody diarrhea [2,3,4]. Mastitis is a common disease in dairy cows. It significantly reduces milk yield and quality if not addressed promptly and effectively, leading to considerable economic losses [5,6]. Mastitis is induced by a variety of factors, including bacteria, fungi, viruses, and other microorganisms and environmental factors, of which *E. coli* is one of the most frequently isolated pathogens in bovine intramammary infections [7,8]. *E. coli* is not only associated with subclinical and clinical mastitis in cows, but it can also transmit antimicrobial resistance (AMR) to humans through the consumption of inadequately heated milk or milk products [9,10]. Risk factors associated with *E*. *coli* contamination include the type of milk container, mammary gland cleaning practices, and farm hygiene management. Cow’s milk can act as a reservoir for antibiotic-resistant enteropathogenic *E. coli*, posing health risks to both animals and humans [11]. As a zoonotic pathogen, *E. coli* is highly transmissible and can rapidly acquire drug resistance, causing clinical diseases across a wide range of hosts, including humans and animals of varying ages [12]. Currently, *E. coli* ranks among the top three bacterial diseases affecting agriculture globally.

For decades, controlling *E. coli* infections in livestock has heavily relied on antibiotic therapy with β-lactams, fluoroquinolones, methotrexate-sulfamethoxazole complexes, and tetracyclines [13,14]. These antibiotics are not only employed for treating infections of *E*. *coli*-related diseases in veterinary and human medicine but are also used prophylactically and as antimicrobial growth promoters in livestock feed in many countries [15]. However, the increasing reliance on antibiotics has driven the rapid development of AMR, which has become a significant global public health challenge. *E. coli* has developed resistance to many antibiotics, which is not confined to specific regions but is, rather, a global phenomenon. AMR arises through horizontal transfer between bacteria or through mutations that develop under selective pressure [16]. The rapidly rising resistance of *E. coli* to multiple antibiotics in human medicine and animal husbandry has been widely documented globally [17]. The inappropriate use of antibiotics in veterinary practice, such as overdosing, improper compounding, or incomplete treatment regimens, exacerbates the development of AMR of *E. coli*. In addition, constant exposure to different antimicrobial drugs fosters the transfer of resistant plasmids, leading to multidrug-resistant (MDR) *E. coli* [18]. For instance, Mwasinga et al. [19] found that 51.2% (214/418) of *E. coli* isolates from raw milk samples were MDR in the Namwala district in Zambia. Similarly, a study by Tripathi et al. [20] showed that 89.8% (575/640) of *E. coli* isolates from raw milk samples were resistant to more than two antimicrobial agents. The spread of MDR bacteria is a considerable challenge to human health and the livestock industry [21,22].

Given the critical role of *E. coli* in mastitis, selecting appropriate antibiotics is essential for effective treatment [23]. Many countries, including India [24], the Kwara State in Nigeria [25], Ethiopia [26], the North Sinai Governorate in Egypt [27], and Romania [28], have conducted studies on the genotypes of AMR *E. coli* in raw milk. While a previous study investigated the distribution of *E. coli* in raw milk from northern China [29], a nationwide study of all regions is needed to provide a comprehensive understanding of AMR *E*. *coli* in China. The presence of pathogenic *E. coli* in raw milk not only facilitates the transmission of antibiotic-resistance genes to the human gut but also complicates treatment approaches. Monitoring antibiotic-resistance profiles is crucial to devise effective therapeutic strategies. Continuous surveillance of the resistance gene patterns of AMR *E. coli* will help to assess the risks associated with *E. coli* contamination of raw milk. Therefore, the aim of this study was to monitor the prevalence and trends of antibiotic resistance in *E. coli* and provide scientific evidence for developing effective control strategies, promoting the rational use of antibiotics in clinical practice, and protecting public health.

## 2. Results

### 2.1. Prevalence of E. coli

Of 81 *E. coli* isolates (23.1%) from 350 samples, 6 (13.6%) were from 44 samples in Northeast China, 5 (11.6%) from 43 samples in North China, 4 (11.1%) from 36 samples in South China, 14 (33.3%) from 42 samples in East China, 50 (29.2%) from 171 samples in Northwest China, and 2 (14.3%) from 14 samples in Southwest China. There was a significant difference in the *E. coli* isolation rates among the six regions (*p* < 0.01).

### 2.2. Antimicrobial Susceptibility Testing

Antimicrobial susceptibility testing showed that 81 *E. coli* isolates from raw milk samples exhibited significant and variable resistance to 14 antimicrobials (Table 1). Notably, phenotypic resistance was observed for 13 (92.9%) of the 14 tested antimicrobials, with the exception of meropenem. Isolates showed the highest resistance to cephalothin (63.0%), with no significant differences observed among the six regions (*p* > 0.05). The resistance rates of *E. coli* isolates for other antimicrobials ranged from 0.0% to 30.9%, with no resistance to meropenem. The resistance rates of the *E. coli* isolates to kanamycin, gentamicin, doxycycline, fosfenicol, doxorubicin, ciprofloxacin, and cotrimoxazole were low at 2.5%, 6.2%, 8.7%, 2.5%, 7.4%, 6.2%, and 5.0%, respectively.

In Northeast China, isolates had the highest rate of resistance to cefotiophene (75.0%), followed by amoxicillin/clavulanic acid and polymyxin E (33.3%), but were sensitive to meropenem, kanamycin, gentamicin, tetracycline, doxycycline, florfenicol, ciprofloxacin, sulfisoxazole, and cotrimoxazole. Isolates from North China had the highest rate of resistance to cefthiophene (60.0%), with sensitivity to amoxicillin/clavulanic acid, meropenem, kanamycin, gentamycin, tetracycline, doxycycline, flucytosine, ciprofloxacin, ciprofloxacin, and cotrimoxazole. Isolates from South China had the highest rate of resistance to cephalothin (50.0%), while all isolates were susceptible to amoxicillin/clavulanic acid, meropenem, kanamycin, gentamicin, tetracycline, doxycycline, florfenicol, polymyxin E, ciprofloxacin, and cotrimoxazole. Isolates from East China had the highest rate of resistance to cefotaxime (71.4%), with sensitivity to meropenem, florfenicol, and polymyxin E. Isolates from Northwest China had the highest rate of resistance to cefthiophene (62.0%) and sensitivity to meropenem. Isolates from Southwest China were resistant only to cefthiophene and sensitive to all other tested antibiotics.

### 2.3. Multidrug Resistance

Multidrug resistance is defined as resistance to at least three or more antimicrobial classes. Of the 81 *E. coli* isolates, 59 (72.8%) were resistant to at least one antibiotic and 27 (33.3%) were resistant to more than three. The MIC distribution of the 27 MDR isolates is shown in Figure 1. The MDR isolates were most commonly resistant to ampicillin, amoxicillin/clavulanic acid, cephalosporins, ceftiofur, tetracyclines, and sulfisoxazole.

### 2.4. Screening Antibiotic Resistance Genes

In total, 9 antibiotic resistance genes were identified among the 27 MDR *E. coli* isolates, which included β-lactam resistance genes (*blaTEM*, *blaSHV*, *blaPSE*, *blaOXA*, *blaCTX-M*), tetracycline resistance genes (*tetA* and *tetB*), and sulfonamide resistance genes (*sul1* and *sul2*). The blaTEM β-lactam resistance gene was the most prevalent, with a detection rate of 92.6% among the 27 MDR isolates. The detection frequencies of *blaSHV*, *blaPSE*, *blaOXA*, *blaCTX-M*, *tetA*, *tetB*, *sul1*, and *sul2* were 25.9%, 14.8%, 37.0%, 37.0%, 37.0%, 29.6%, 11.1%, and 25.9%, respectively (Figure 2).

The *E. coli* isolates exhibited varied resistance phenotypes to different antimicrobial drugs. A comparison of phenotypic resistance and the corresponding resistance genes of *E. coli* isolates from raw milk samples showed that isolates resistant to β-lactams, tetracyclines, and sulfonamides also harbored the respective resistance genes (*blaTEM*, *blaSHV*, *blaPSE*, *blaOXA*, *blaCTX-M*, *tetA*, *tetB*, *sul1*, and *sul2*) (Table 2 and Figure 2). The resistance rate of the isolates to β-lactam antibiotics was 100% and most of the isolates were able to amplify the corresponding resistance genes. The different combination patterns of resistance genes detected in this study are shown in Table 2. All of the isolates carried at least 1 resistance gene and 19 unique gene combinations were detected. The most frequent patterns involved isolates carrying three genes (*n* = 7, 25.9%), followed by one gene (*n* = 6, 22.2%), five genes (*n* = 5, 18.5%), two genes (*n* = 4, 14.8%), four genes (*n* = 3, 11.1%), and six genes (*n* = 2, 7.4%). The most common gene combinations were the single-gene pattern of *blaTEM* (*n* = 5, 18.5%) and the three-gene pattern of *blaTEM-blaPSE-blaOXA* (*n* = 3, 11.1%).

## 3. Discussion

Mastitis is a devastating disease in dairy farms that not only reduces milk production and quality but also increases culling rates among lactating cows. *E. coli*, a widespread environmental pathogen, is a key contributor to mastitis [30,31]. In addition, *E*. *coli* is a zoonotic pathogen that is particularly capable of acquiring AMR, which may have serious implications for animal food safety and human health [32]. At present, treatment of *E. coli* infections primarily relies on antimicrobial agents [33]. However, the prevalence of drug resistance continues to escalate, particularly MDR bacteria, often linked to the overuse or inappropriate application of antibiotics [34]. The rise in drug resistance affects livestock production and poses a threat to human health, complicating treatment protocols and potentially leading to increased healthcare costs and treatment failures. Continuous monitoring and comprehensive research into MDR *E. coli* are essential for developing effective infection control strategies and promoting responsible antibiotic use. Therefore, in this study, we analyzed the prevalent drug-resistant phenotypes and genotypes of 81 *E. coli* isolates from 350 raw milk samples collected from healthy cows at large-scale farms in 16 provinces in six regions of China. Our findings enhance the understanding of the prevalence of *E. coli* in raw milk and associated drug resistance risks, thereby supporting efforts to rationalize antimicrobial use, minimize misuse in animal husbandry, and protect public health.

The results indicate a prevalence of *E. coli* in raw milk of 23.1% (81/350). This detection rate was lower than in previous reports of 70.4% in Indonesia [35], 70% in Bangladesh [36], 45% in Northern China [37], and 41.2% in Tennessee, USA [38]. In contrast, our findings were higher than the 11.8% incidence of *E. coli* in raw milk in Kenya [39]. Moreover, our results are similar to those of Awadallah et al. [40] who reported that 22.4% of raw milk samples were positive for *E. coli* in the Sharkia Governorate area. There were no significant differences in the prevalence of *E. coli* or the resistance patterns of the isolates depending on the source of the samples. Overall, the high isolation rate of *E. coli* in the raw milk samples of this study indicates the risk of contamination during the production process, suggesting that the frequent occurrence of mastitis in dairy cows might be caused by irregular operation during milking and poor environmental hygiene conditions.

Antibiotics are commonly administered for the treatment of mastitis, although this practice contributes to the emergence of MDR isolates [41]. In China, approximately 6000 tons of veterinary antibiotics are used annually, mostly as feed additives, including β-lactams and tetracyclines [42]. This widespread use indicates multiple pathways for the development of resistance genes, which can be transferred between pathogens.

In the present study, drug sensitivity testing of 81 *E. coli* isolates revealed varying levels of resistance to several antimicrobial drugs, except for meropenem. Among the isolates, 63.0% were resistant to cephalothin and all were sensitive to meropenem. These findings are similar to a study by Ibrahim et al. [43] that suggested 97.1% of *E. coli* isolates were resistant to ampicillin and 71.4% were resistant to compounded amoxicillin, cefotaxime, ceftazidime, and ceftiofur. A study conducted in Bangladesh by Rana et al. [44] revealed that the highest resistance rates were observed for ampicillin and tetracycline (100%), followed by amoxicillin (79.17%), ceftazidime (62.5%), streptomycin (58.53%), and gentamicin (60%). In contrast, the bacteria were most sensitive to vancomycin, ciprofloxacin, and meropenem. A phenotypic study of *E. coli* isolates from raw milk samples showed that 82.25% of isolates were resistant to ampicillin and 50% to sulfamethoxazole [45]. Antibiotic susceptibility of *E. coli* is crucial for the selection of appropriate antibiotics for the treatment of mastitis. These studies have pointed out that *E. coli* exhibits a high degree of resistance to β-lactams, tetracyclines, and sulfisoxazoles, consistent with the results of the present study. This phenomenon may be due to the widespread use of β-lactams, tetracyclines, and sulfisoxazoles against *E. coli* infections in clinical therapy, suggesting that antibiotics should be used rationally in veterinary clinics.

In recent years, MDR *E. coli* has become an increasingly serious public health concern worldwide [46]. In the present study, 9 (33.3%) of 27 *E. coli* isolates exhibited multidrug resistance, consistent with a report by Lan et al. [47] that suggested 34.80% of *E. coli* isolates exhibited multidrug resistance in some parts of China. In comparison, the multi-resistance rate in the present study was significantly lower than the 84.20% rate reported by Bag et al. [48] in Bangladeshi dairy cows and the 100% rate reported by Eldesoukey et al. [49]. The high prevalence of *E. coli* infection and the increasing prevalence of multidrug resistance poses a significant risk to public health and food safety. Differences in multidrug resistance rates among regions may be due to differences in the type and amount of antibiotics used on different ranches. Therefore, rational control of antibiotic use is key to preventing the development of multidrug resistance. The emergence of MDR *E. coli* is related to the irrational use of antimicrobial drugs and the drug-resistance genes carried by *E. coli*. β-lactamase is the main cause of bacterial resistance to β-lactam antibiotics, which is a bacterial enzyme that can inactivate β-lactam antibiotics through hydrolysis, thus rendering these drugs ineffective [50]. In the present study, all 27 *E. coli* isolates carried β-lactamase-encoding genes and the detection rates of *blaPSE*, *blaSHV*, *blaOXA*, *blaCTX-M*, and *blaTEM* were 14.8%, 25.9%, 37.0%, 37.0%, and 92.6%, respectively. Among the genes, *blaTEM* had the highest detection rate, which was similar to the *blaTEM* gene (*n* = 69, 83.1%) reported by Yu et al. [51], who have shown that treatment of mastitis with cephalosporins increased the proportion of blaTEM in milk samples during the withdrawal period, thus likely contributing to the high detection rate of *blaTEM* [52]. The *blaTEM* and *blaOXA* genes were detected in 10 *E. coli* isolates, *blaTEM* and *blaCTX-M* in 8 isolates, and *blaTEM* and *blaSHV* in 6 isolates, suggesting a high detection rate of β-lactamase resistance genes. Thus, *E. coli* can serve as a reservoir of antibiotic resistance and possible gene transfer to other pathogenic species. In general, *tetA* and *tetB* are the most prevalent tetracycline resistance genes in *E. coli* of animal origin [53]. These genes are components of the small nonconjugative transposons Tn1721 (*tetA*) [54] and Tn10 (*tetB*) [55], which are frequently integrated into conjugative and nonconjugative plasmids. Among the 27 MDR isolates in this study, the detection rates of *tetA* and *tetB* were 37.0% and 29.6%, respectively, consistent with several previous studies [56,57]. Sulfonamide resistance mediated by the *sul* gene has become globally prevalent, especially in *E. coli* in food and companion animals, where sulfonamide antibiotic resistance is severe [58]. In the present study, the detection rates of *sul1* and *sul2* were 11.1% and 25.9%, respectively, consistent with previously reported data on *E. coli* isolated from food and animals [59,60]. In total, 19 different resistance gene combination patterns were identified in this study, with 7 (25.9%) *E. coli* isolates carrying three resistance genes. In this study, the isolates were resistant to β-lactams, tetracyclines, and sulfonamides, consistent with the detected resistance genes (*blaTEM*, *blaSHV*, *blaPSE*, *blaOXA*, *blaCTX-M*, *tetA*, *tetB*, *sul1*, and *sul2*). The resistance rate of the isolates to β-lactams was 100% and most of the isolates were able to amplify the corresponding resistance genes. The presence of these resistance genes is key to the development of drug resistance in *E. coli*.

In summary, *E. coli*, the main pathogen causing mastitis in dairy cows, has demonstrated resistance to a wide range of antibiotics. *E. coli* is known to be a very efficient reservoir of antibiotic-resistance genes and can transfer these genes to other pathogens. Therefore, analyzing *E. coli* isolates for resistance and related resistance genes is essential for screening suitable antimicrobial drugs.

## 4. Materials and Methods

### 4.1. Sample Collection

Overall, 350 raw milk samples were collected from large-scale farms across six regions in China (Northeast, North, South, East, Northwest, and Southwest), which encompassed 16 provinces and municipalities, including autonomous regions (Beijing, Tianjin, Heilongjiang, Inner Mongolia, Gansu, Shaanxi, Hebei, Shandong, Jiangsu, Zhejiang, Fujian, Guangdong, Chongqing, Yunnan, Guizhou, and Xinjiang). In total, 44, 43, 36, 42, 171, and 14 raw milk samples were collected in Northeast, South, East, Northwest, and Southwest China, respectively. The raw milk samples were collected from the top, middle, and bottom of bulk tanks, mixed well, and then transferred into sterile bottles and immediately transported to our laboratory at 4 °C.

### 4.2. Isolation and Characterization of E. coli

An aliquot of 25 mL from each milk sample was mixed with 225 mL of trypticase soy broth and then incubated at 37 °C for 16 h with shaking to detect *E. coli*. Following incubation, the samples were incubated on MacConkey agar plates (Beijing Luqiao Technology Co., Ltd., Beijing, China) for 18–24 h at 37 °C. Presumptive *E. coli* colonies, which appeared small and pink, with smooth, moist surfaces and neat edges, were selected for Gram staining and biochemical identification. Gram-negative *E. coli* isolates appeared microscopically as red, short rods, positive for methyl red, and negative for Voges–Proskauer and citric acid. Also, the isolates were cultured in 3 mL of trypsinized soy broth at 37 °C for 18–24 h whilst being shaken for DNA extraction, which was conducted with the TIANamp Bacteria DNA Kit (Tiangen Biotech (Beijing) Co., Ltd., Beijing, China) in accordance with the manufacturer’s instructions. The extracted genomic DNA samples were stored at −20 °C until further use.

### 4.3. PCR Amplification

PCR amplification was performed by Beijing Bio-Tech Co., Ltd. (Beijing, China) using 16S rRNA universal primers and specific primers for the *phoA* gene (Appendix A). Each 30 μL PCR reaction mixture consisted of 15 μL of 2×Taq PCR Master Mix, 1 μL each of the forward and reverse primers (10 pmol/μL), 2 μL of DNA template, and 11 μL of double-distilled H_2_O. The PCR amplification program included an initial denaturation step at 94 °C for 4 min, followed by 30 cycles of denaturation at 94 °C for 30 s, annealing at 57 °C for 30 s, and extension at 72 °C for 90 s. The PCR amplification products were analyzed by electrophoresis on a 1.0% agarose gel and the amplicons were sequenced by Beijing Dingguo Changsheng Biotechnology Co., Ltd. (Beijing, China). The obtained sequences were assessed against the National Center for Biotechnology Information database (https://www.ncbi.nlm.nih.gov/ (accessed on 1 March 2024)) using the Basic Local Alignment Search Tool (https://blast.ncbi.nlm.nih.gov/Blast.cgi (accessed on 1 March 2024)).

### 4.4. Antimicrobial Susceptibility Patterns

Antimicrobial susceptibility testing for 14 antimicrobial agents was conducted using the broth dilution method as recommended by the American Committee for Clinical and Laboratory Standardization (CLSI, 2024). The bacterial suspensions were adjusted to a turbidity of 0.5 McFarland units using a turbidimeter and inoculated into 96-well plates containing different concentrations of the following antimicrobial agents: ampicillin (0.25–128 mg/L), amoxicillin/clavulanic acid (0.25/0.12–128/64 mg/L), cephalothin (0.25–128 mg/L), ceftiofur (0.25–128 mg/L), meropenem (0.25–128 mg/L), kanamycin (0.25–128 mg/L), gentamicin (0.25–128 mg/L), tetracycline (0.25–128 mg/L), doxycycline (0.25–128 mg/L), florfenicol (0.25–128 mg/L), polymyxin E (0.25–128 mg/L), ciprofloxacin (0.25–128 mg/L), sulfisoxazole (2–1024 mg/L), and sulfamethoxazole (0.12/2.4–64/1216 mg/L) (YNK (Tianjin) Biotechnology Co., Ltd., Tianjin, China). The plates were incubated at 35 °C for 18–20 h. The minimum inhibitory concentration (MIC), defined as the lowest concentration of an antimicrobial agent that inhibited visible bacterial growth, was recorded. The interpretation of the results was based on the guidelines of CLSI (2024) (Appendix A).

### 4.5. Detection of Drug Resistance Genes

Five genes related to β-lactamase resistance (*blaTEM*, *blaSHV*, *blaPSE*, *blaOXA*, and *blaCTX-M*), two related to tetracycline resistance (*tetA* and *tetB*), and two related to sulfisoxazole resistance (*sul1* and *sul2*) of the *E*. *coli* isolates were detected by multiplex PCR (Appendix A). The PCR amplification conditions included an initial denaturation step at 94 °C for 5 min, followed by 30 cycles of denaturation at 94 °C for 45 s, annealing at different temperatures for 30 s, extension at 72 °C for 45 s, and extension at 72 °C for 10 min. *E. coli* strain ATCC 25922 served as a positive control in each run. Supplementary material provides specific details of sample numbers (Appendix A).

### 4.6. Statistical Analysis

All data were recorded and organized with Excel 2019 software (Microsoft Corporation, Redmond, WA, USA). Statistical comparisons between groups, including prevalence rates, AMR phenotypes, and the distribution of resistance genes, were performed using one-way analysis of variance and the least significant difference multiple comparisons with IBM SPSS Statistics for Windows (version 20.0; IBM Corporation, Armonk, NY, USA). A probability (*p*) value < 0.05 was considered statistically significant.

## 5. Conclusions

In this study, *E. coli* isolated from raw milk produced in China was characterized for the first time. The results showed that the prevalence of *E. coli* in raw milk in China was as high as 23.1% and the multidrug resistance rate was 33.3%. Therefore, the in-depth monitoring of the prevalence of *E. coli* and antibiotic resistance is particularly important for the development of effective preventive and control measures, and to promote the rational clinical use of antibiotics. Further studies are warranted to assess the mechanism employed by *E. coli* to acquire, transfer, and transmit multidrug resistance genes.

## Figures and Tables

**Figure 1 ijms-26-00454-f001:**
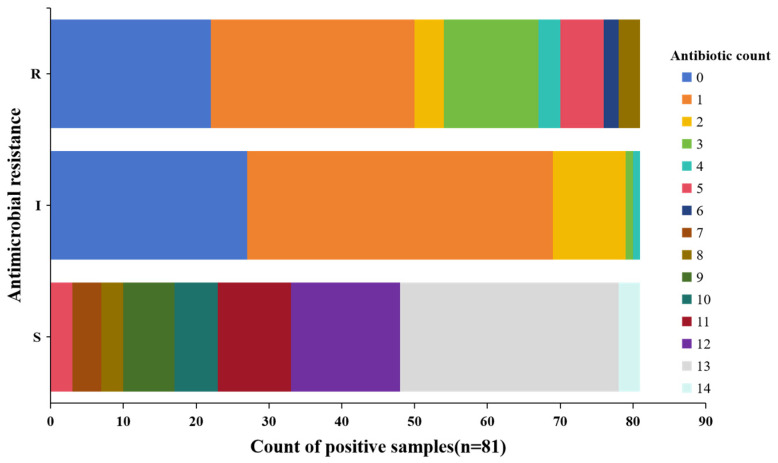
Patterns of multidrug resistance for Raw Bovine Milk-Derived *E. coli* based on resistance classifications as susceptible (S), resistant (R), or intermediate (I).

**Figure 2 ijms-26-00454-f002:**
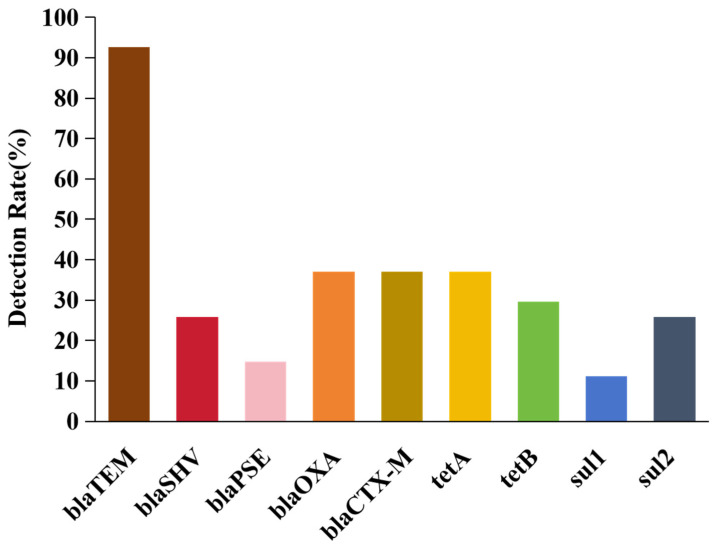
Detection rate of drug resistance genes *blaTEM* (92.6%), *blaSHV* (25.9%), *blaPSE* (14.8%), *blaOXA* (37.0%), *blaCTX-M* (37.0%), *tetA* (37.0%), *tetB* (29.6%), *sul1* (11.1%), and *sul2* (25.9%). The resistance genes of *blaTEM*, *blaSHV*, *blaPSE*, *blaOXA*, and *blaCTX-M* belong to beta-lactams; the *tetA* and *tetB* genes belong to tetracyclines; and the *sul1* and *sul2* genes belong to sulfonamides.

**Table 1 ijms-26-00454-t001:** Antibiotic resistance of isolated *E. coli* strains.

Name of Antibacterial Drugs	No. (%) of Positive Strains						
	Northeast China (*n* = 6)	North China (*n* = 5)	South China (*n* = 4)	East China (*n* = 14)	Northwest China (*n* = 50)	Southwest China (*n* = 2)	Total (*n* = 81)
Ampicillin	1 (16.7)	2 (40.0)	1 (25.0)	7 (50.0)	14 (28.0)	0 (0.0)	25 (30.9)
Amoxicillin/Clavulanic acid	2 (33.3)	0 (0.0)	0 (0.0)	4 (28.6)	3 (6.0)	0 (0.0)	9 (12.4)
Cephalothin	4 (75.0)	3 (60.0)	2 (50.0)	10 (71.4)	31 (62.0)	1 (50.0)	51 (63.0)
Ceftiofur	1 (16.7)	2 (40.0)	1 (25.0)	1 (7.1)	9 (18.0)	0 (0.0)	14 (17.3)
Meropenem	0 (0.0)	0 (0.0)	0 (0.0)	0 (0.0)	0 (0.0)	0 (0.0)	0 (0.0)
Kanamycin	0 (0.0)	0 (0.0)	0 (0.0)	1 (7.1)	1 (2.0)	0 (0.0)	2 (2.5)
Gentamicin	0 (0.0)	0 (0.0)	0 (0.0)	3 (21.4)	2 (4.0)	0 (0.0)	5 (6.2)
Tetracycline	0 (0.0)	0 (0.0)	0 (0.0)	2 (14.3)	9 (18.0)	0 (0.0)	11 (13.6)
Doxycycline	0 (0.0)	0 (0.0)	0 (0.0)	1 (7.1)	6 (12.0)	0 (0.0)	7 (8.7)
Florfenicol	0 (0.0)	0 (0.0)	0 (0.0)	0 (0.0)	2 (4.0)	0 (0.0)	2 (2.5)
Polymyxin E	2 (33.3)	1 (20.0)	0 (0.0)	0 (0.0)	3 (6.0)	0 (0.0)	6 (7.4)
Ciprofloxacin	0 (0.0)	0 (0.0)	0 (0.0)	1 (7.1)	4 (8.0)	0 (0.0)	5 (6.2)
Sulfisoxazole	0 (0.0)	1 (20.0)	1 (25.0)	6 (42.9)	5 (10.0)	0 (0.0)	13 (16.1)
Sulfamethoxazole	0 (0.0)	0 (0.0)	0 (0.0)	1 (7.1)	3 (6.0)	0 (0.0)	4 (5.0)

**Table 2 ijms-26-00454-t002:** Main resistance patterns of 27 multidrug-resistant strains of *E. coli* isolated from raw milk.

No. of Resistance Genes	Combination Patterns of Resistance Genes	No. of Resistance Gene Combination Patterns	No. of *E. coli* Isolates (%) (*n* = 27)
1	*blaTEM*	5	6 (22.2)
	*blaCTX-M*	1	
2	*blaTEM-blaOXA*	2	4 (14.8)
	*blaSHV-blaCTX-M*	1	
	*blaTEM-blaCTX-M*	1	
3	*blaTEM-blaPSE-blaOXA*	3	7 (25.9)
	*blaTEM-blaSHV-blaCTX-M*	2	
	*blaTEM-tetA-tetB*	1	
	*blaTEM-blaCTX-M-tetA*	1	
4	*blaTEM-blaSHV-blaPSE-blaOXA*	1	3 (11.1)
	*blaTEM-blaSHV-blaOXA-sul1*	1	
	*blaTEM-tetA-tetB-sul2*	1	
5	*blaTEM-blaCTX-M-tetA-sul1-sul2*	1	5 (18.5)
	*blaTEM-blaSHV-tetA-tetB-sul2*	1	
	*blaTEM-blaOXA-tetA-tetB-sul2*	1	
	*blaTEM-blaOXA-blaCTX-M-tetA-tetB*	1	
	*blaTEM-tetA-tetB-sul1-sul2*	1	
6	*blaTEM-blaOXA-blaCTX-M-tetA-tetB-sul2*	1	2 (7.4)
	*blaTEM-blaSHV-blaCTX-M-tetA-tetB-sul2*	1	

## Data Availability

Data are available in Appendix A.

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
