# Peer review of "Prevalent and Drug-Resistant Phenotypes and Genotypes of Escherichia coli Isolated from Healthy Cow’s Milk of Large-Scale Dairy Farms in China"

_ijms, 2025, doi:10.3390/ijms26020454_

Round 1
Reviewer 1 Report
Comments and Suggestions for Authors
The authors investigated the prevalence and trends of antibiotic resistance in Escherichia coli isolated from raw milk in China.
While the data presented are valuable, the methodology employed is limited, relying on outdated techniques. Although the results are accurate, they are constrained by the absence of whole genome sequencing for the strains analyzed.
To provide a more comprehensive analysis, it is recommended that at least extended-spectrum beta-lactamase (ESBL)–producing and colistin-resistant bacteria be examined using WGS. This approach would allow for the prediction of the resistome, determination of sequence types, and identification of plasmid types, thereby enhancing the study’s overall robustness and applicability.
Author Response
Comments 1: The authors investigated the prevalence and trends of antibiotic resistance in Escherichia coli isolated from raw milk in China. While the data presented are valuable, the methodology employed is limited, relying on outdated techniques. Although the results are accurate, they are constrained by the absence of whole genome sequencing for the strains analyzed. To provide a more comprehensive analysis, it is recommended that at least extended-spectrum beta-lactamase (ESBL)–producing and colistin-resistant bacteria be examined using WGS. This approach would allow for the prediction of the resistome, determination of sequence types, and identification of plasmid types, thereby enhancing the study’s overall robustness and applicability. |
Response 1: Thanks to the reviewers for these valuable comments and suggestions, Escherichia coli in raw milk is an important hygienic indicator for evaluating raw milk and dairy products, but the relative lack of data in China affects our comprehensive understanding of its potential risk. In this study, we conducted an in-depth analysis of Escherichia coli strains isolated from raw milk of Chinese dairy cows, focusing on the susceptibility of these strains to commonly used antimicrobial drugs. The findings showed that some strains have demonstrated resistance to multiple antimicrobial drugs. This finding emphasises the importance of strengthening the monitoring and control of E. coli in raw milk to ensure public health safety. Currently, our team is building on this study to continue exploring the correlation between the resistance genes carried by E. coli and the mechanism of resistance. This will provide new insights into our understanding of the evolution of drug-resistant E. coli and provide a scientific basis for the development of targeted and effective antibiotic strategies. Finally, we would like to thank the reviewers again for their valuable suggestions, which provide an important reference value for the improvement of the subsequent experiments. We will seriously consider these suggestions and further improve our research work. |

Reviewer 2 Report
Comments and Suggestions for Authors
Review
Work reported in the manuscript “Prevalence and drug-resistant phenotypes and genotypes of Escherichia coliisolated from healthy cow's milk of large-scale dairy farms in China” investigates the prevalence of antibiotic resistance in strains isolated from healthy cow’s milk samples in different regions of China.
Please consider these suggestions for improving this manuscript:
Minor
Abstract
Line 19. “prevalence” instead of “contamination”
Introduction
Lines 71, 73. When you mention the authors of a study you should write the surname of the first author followed by “et al.,”. For example “Mwasinga, et al.,” instead of “Wizaso Mwasinga et al.”
Results
Line 124. Table 1: “Antibiotic resistance of isolated E.coli strains” instead of “Antibiotic resistance of strains”
Line 129. “is” instead of “was”
Figure 1. The authors should consider changing the colors of antibiotics numbers 1 and 13, as the colors presented are not easily distinguishable.
Discussion
Line 166. Please provide some reference for “Dairy cow mastitis is a devastating disease in dairy farms that not only reduces milk production and quality, but also increases culling rates among lactating cows.”
Line 193. “study” instead of “trial”
Lines 203-205. “81 isolates of E. coli revealing varying levels of resistance to several antimicrobial drugs, except for meropenem.” Instead of “81 isolates of E. coli and the results showed that these E. coli isolates showed varying degrees of resistance to some of the antimicrobial drugs, with the exception of meropenem.”
Line 207. “to” instead of “as”
Line 207. Ibrahim, et al.
Line 209. Rana, et al.
Lines 209-213. Should be written something like: “Rana et al.[43], investigated the drug resistance of E. coli in raw milk from Bangladesh. Their findings revealed that the highest resistance rates were observed for ampicillin and tetracycline (100%), followed by amoxicillin (79.17%), ceftazidime (62.5%), streptomycin (58.53%), and gentamicin (60%). In contrast, the bacteria were most sensitive to vancomycin, ciprofloxacin, and meropenem.”
Line 215. “and” instead of “,”
Line 226, 229, 230. “Lan, et al.”, “Bag, et al.”, “Eldesoukey, et al.”
Line 240. “in the 27…” instead of “in 27…”
Line 240. E.coli should be in italics
Line 241. “…92,6%, respectively.”
Material and Methods
4.2. Isolation and characterization of E. coli. Authors should provide more information on the biochemical tests they used for E. coli identification.
4.3. PCR Amplification. Authors should provide the details of the kit they used for the PCR
Line 307. “amplicons” instead of “amplified sequences”
Comments on the Quality of English Language
I believe this manuscript would benefit from being reviewed by a native English speaker.
Author Response
Comments 1: Work reported in the manuscript “Prevalence and drug-resistant phenotypes and genotypes of Escherichia coli isolated from healthy cow's milk of large-scale dairy farms in China” investigates the prevalence of antibiotic resistance in strains isolated from healthy cow’s milk samples in different regions of China.
Please consider these suggestions for improving this manuscript: Minor Abstract Line 19. “prevalence” instead of “contamination” Introduction Lines 71, 73. When you mention the authors of a study you should write the surname of the first author followed by “et al.,”. For example “Mwasinga, et al.,” instead of “Wizaso Mwasinga et al.” Results Line 124. Table 1: “Antibiotic resistance of isolated E.coli strains” instead of “Antibiotic resistance of strains” Line 129. “is” instead of “was” Figure 1. The authors should consider changing the colors of antibiotics numbers 1 and 13, as the colors presented are not easily distinguishable. Discussion Line 166. Please provide some reference for “Dairy cow mastitis is a devastating disease in dairy farms that not only reduces milk production and quality, but also increases culling rates among lactating cows.” Line 193. “study” instead of “trial” Lines 203-205. “81 isolates of E. coli revealing varying levels of resistance to several antimicrobial drugs, except for meropenem.” Instead of “81 isolates of E. coli and the results showed that these E. coli isolates showed varying degrees of resistance to some of the antimicrobial drugs, with the exception of meropenem.” Line 207. “to” instead of “as” Line 207. Ibrahim, et al. Line 209. Rana, et al. Lines 209-213. Should be written something like: “Rana et al.[43], investigated the drug resistance of E. coli in raw milk from Bangladesh. Their findings revealed that the highest resistance rates were observed for ampicillin and tetracycline (100%), followed by amoxicillin (79.17%), ceftazidime (62.5%), streptomycin (58.53%), and gentamicin (60%). In contrast, the bacteria were most sensitive to vancomycin, ciprofloxacin, and meropenem.” Line 215. “and” instead of “,” Line 226, 229, 230. “Lan, et al.”, “Bag, et al.”, “Eldesoukey, et al.” Line 240. “in the 27…” instead of “in 27…” Line 240. E.coli should be in italics Line 241. “…92,6%, respectively.” Material and Methods 4.2. Isolation and characterization of E. coli. Authors should provide more information on the biochemical tests they used for E. coli identification. 4.3. PCR Amplification. Authors should provide the details of the kit they used for the PCR Line 307. “amplicons” instead of “amplified sequences”
|
Response 2: 1. Line 19. “prevalence” instead of “contamination” We sincerely thank the reviewer for careful reading. As suggested by the reviewer, we have corrected the “contamination” into “prevalence”, as detailed in line 19.
2. Lines 71, 73. When you mention the authors of a study you should write the surname of the first author followed by “et al.,”. For example “Mwasinga, et al.,” instead of “Wizaso Mwasinga et al.” We sincerely thank the reviewer for careful reading. As suggested by the reviewer, we have made changes to follow the correct citation format, as detailed in line 72,73.
3. Line 124. Table 1: “Antibiotic resistance of isolated E.coli strains” instead of “Antibiotic resistance of strains” We sincerely thank the reviewer for careful reading. As suggested by the reviewer, we have corrected the “Antibiotic resistance of strains” into “Antibiotic resistance of isolated E.coli strains”, as detailed in line 124.
4. Line 129. “is” instead of “was” We sincerely thank the reviewer for careful reading. As suggested by the reviewer, we have corrected the “was” into “is”, as detailed in line 129.
5. Figure 1. The authors should consider changing the colors of antibiotics numbers 1 and 13, as the colors presented are not easily distinguishable. We sincerely thank the reviewer for careful reading. As suggested by the reviewer, we have adjusted the colours in Figure 1, as detailed in Figure 1
6. Line 166. Please provide some reference for “Dairy cow mastitis is a devastating disease in dairy farms that not only reduces milk production and quality, but also increases culling rates among lactating cows.” We sincerely thank the reviewer for careful reading. As suggested by the reviewer, we have added two references the “31.Goulart DB, Mellata M. Escherichia coli Mastitis in Dairy Cattle: Etiology, Diagnosis, and Treatment Challenges. Front Microbiol. 2022;13:928346. doi: 10.3389/fmicb.2022.928346” and “32.Asrat Asfaw. Prevalence and Potential Risk Factors of Bovine Clinical Mastitis in Bonke District, Gamo Zone, Southern Ethiopia. OMO International Journal of Sciences. 2023;6 (1):1-11. doi:10.59122/13462rw”.
7. Line 193. “study” instead of “trial” We sincerely thank the reviewer for careful reading. As suggested by the reviewer, we have corrected the “trial” into “study”, as detailed in line 189.
8. Lines 203-205. “81 isolates of E. coli revealing varying levels of resistance to several antimicrobial drugs, except for meropenem.” Instead of “81 isolates of E. coli and the results showed that these E. coli isolates showed varying degrees of resistance to some of the antimicrobial drugs, with the exception of meropenem.” We sincerely thank the reviewer for careful reading. As suggested by the reviewer, we have corrected the “81 isolates of E. coli and the results showed that these E. coli isolates showed varying degrees of resistance to some of the antimicrobial drugs, with the exception of meropenem.” into “81 isolates of E. coli revealing varying levels of resistance to several antimicrobial drugs, except for meropenem.”, as detailed in line 198,199.
9. Line 207. “to” instead of “as” We sincerely thank the reviewer for careful reading. As suggested by the reviewer, we have corrected the “as” into “to”, as detailed in line 201.
10. Line 207. Ibrahim, et al. We sincerely thank the reviewer for careful reading. As suggested by the reviewer, we have made changes to follow the correct citation format, as detailed in line 201.
11. Line 209. Rana, et al. We sincerely thank the reviewer for careful reading. As suggested by the reviewer, we have made changes to follow the correct citation format, as detailed in line 203.
12. Lines 209-213. Should be written something like: “Rana et al.[43], investigated the drug resistance of E. coli in raw milk from Bangladesh. Their findings revealed that the highest resistance rates were observed for ampicillin and tetracycline (100%), followed by amoxicillin (79.17%), ceftazidime (62.5%), streptomycin (58.53%), and gentamicin (60%). In contrast, the bacteria were most sensitive to vancomycin, ciprofloxacin, and meropenem.” We sincerely thank the reviewer for careful reading. As suggested by the reviewer, we have modified the original text to read as follows“A study conducted in Bangladesh by Rana et al. [45] revealed that the highest resistance rates were observed for ampicillin and tetracycline (100%), followed by amoxicillin (79.17%), ceftazidime (62.5%), streptomycin (58.53%), and gentamicin (60%). In contrast, the bacteria were most sensitive to vancomycin, ciprofloxacin, and meropenem.”, as detailed in line
13. Line 215. “and” instead of “,” We sincerely thank the reviewer for careful reading. As suggested by the reviewer, we have corrected the “ ” into “ ”, as detailed in line 208.
14. Line 226, 229, 230. “Lan, et al.”, “Bag, et al.”, “Eldesoukey, et al.” We sincerely thank the reviewer for careful reading. As suggested by the reviewer, we have made changes to follow the correct citation format, as detailed in line 218,221,222.
15. Line 240. “in the 27…” instead of “in 27…” We sincerely thank the reviewer for careful reading. As suggested by the reviewer, we have corrected the “in 27” into “all 27”, as detailed in line 231.
16. Line 240. E.coli should be in italics We sincerely thank the reviewer for careful reading. As suggested by the reviewer, we have changed “E. coli”to an italicised.
17. Line 241. “…92,6%, respectively.” We sincerely thank the reviewer for careful reading. As suggested by the reviewer, we have corrected the “…92,6%, respectively”, as detailed in line 233.
18. 4.2. Isolation and characterization of E. coli. Authors should provide more information on the biochemical tests they used for E. coli identification. We sincerely thank the reviewer for careful reading. As suggested by the reviewer, we have added specific biochemical results“Gram-negative E. coli isolates appeared microscopically as red, short rods, positive for methyl red, and negative for Voges–Proskauer and citric acid.”, as detailed in line 280-282.
19. 4.3. PCR Amplification. Authors should provide the details of the kit they used for the PCR Line 307. “amplicons” instead of “amplified sequences” We sincerely thank the reviewer for careful reading. As suggested by the reviewer, we have added details of the kits used for PCR and corrected the “amplified sequences” into “amplicons”, as detailed in line 288 and 295.
|

Round 2
Reviewer 1 Report
Comments and Suggestions for Authors/